



# Simulated methane emissions from Arctic ponds are highly sensitive to warming

Zoé Rehder[1,2], Thomas Kleinen[1], Lars Kutzbach[3,4], Victor Stepanenko[5,6,7], Moritz Langer[8,9], and Victor Brovkin[1,4]

[1]Department of the Ocean in the Earth System, Max-Planck-Institute for Meteorology, Hamburg, Germany
[2]International Max Planck Research School on Earth System Modeling, Hamburg, Germany
[3]Institute of Soil Science, Universität Hamburg, Hamburg, Germany
[4]Center for Earth System Research and Sustainability, Universität Hamburg, Hamburg, Germany
[5]Research Computing Center, Moscow State University, Moscow, Russia
[6]Faculty of Geography, Moscow State University, Moscow, Russia
[7]Moscow Center of Fundamental and Applied Mathematics, Moscow, Russia
[8]Alfred Wegener Institute, Helmholtz Centre for Polar and Marine Research, Potsdam, Germany
[9]Department of Earth Sciences, Vrije Universiteit Amsterdam, Amsterdam, the Netherlands

**Correspondence:** Zoé Rehder (zoe.rehder@mpimet.mpg.de)

**Abstract.** We employ a new, process-based model for methane emissions from ponds (MeEP) to investigate the methane-emission response of polygonal-tundra ponds in Northeast Siberia to warming. Small and shallow water bodies such as ponds are vulnerable to warming due to their low thermal inertia compared to larger lakes, and the Arctic is warming at an above-average rate. While ponds are a relevant landscape-scale source of methane under the current climate, the response of pond

methane emissions to warming is uncertain. MeEP differentiates between the three main pond types of the polygonal tundra, ice-wedge, polygonal-center, and merged polygonal ponds. The model resolves the three main pathways of methane emissions – diffusion, ebullition, and plant-mediated transport – at the temporal resolution of one hour, thus capturing daily and seasonal variability of the methane emissions. The model was tuned using chamber measurements resolving the three methane pathways. We perform idealized warming experiments, with increases in the mean annual temperature of 2.5, 5, and $7.5\,^\circ$C on top of a

historical simulation. The simulations reveal an overall increase of $1.33$ g $CH_4$ year$^{-1}$ $^\circ$C$^{-1}$ per square meter of pond area. Under annual temperatures $5\,^\circ$C above present temperatures pond methane emissions are more than three times higher than now. Most of this emission increase is due to the additional substrate provided by the increased net productivity of the vascular plants. Furthermore, plant-mediated transport is the dominating pathway of methane emissions in all simulations. We conclude that vascular plants as a substrate source and efficient methane pathway should be included in future pan-Arctic assessments

of pond methane emissions.

## 1 Introduction

We present results of the first dedicated model for Methane Emissions from Ponds (MeEP). This model has been developed to simulate methane dynamics in Arctic ponds since Arctic landscapes have a high areal coverage of water bodies (Muster et al.,





2017) and ponds are the most numerous among those water bodies (Downing et al., 2006; Polishchuk et al., 2018; Muster et al.,
2019). We define ponds using the Ramsar classification, which uses a size limit of $8 \cdot 10^4$ m$^2$ (Ramsar Convention Secretariat,
2016). We impose the additional condition that the average depth of a pond is less than 2 meters (Lim et al., 2001) which means
that they are almost always well mixed. In our study region, water bodies that are shallow freeze through in winter. Thus, they
do not feature an unfrozen sediment layer throughout the year (talik) (Pienitz et al., 2008; Arp et al., 2012; Surdu et al., 2014).

Ponds are an important component within the Arctic carbon cycle (Abnizova et al., 2012), as they emit carbon dioxide
and, notably, methane (Wik et al., 2016; Holgerson and Raymond, 2016; Beckebanze et al., 2021), which is the greenhouse
gas with the higher warming potential among the two. However, the Arctic is warming rapidly (Chapman and Walsh, 1993;
Bekryaev et al., 2010; Rantanen et al., 2022), which induces a multitude of changes to the permafrost landscape, and to the
embedded ponds, specifically. Ponds are vulnerable to climate change due to their small size and low thermal inertia compared
to lakes. During longer ice-free seasons, more water is lost to evaporation and subsurface runoff (Anderson et al., 2013; Riordan
et al., 2006). So far, Arctic ponds have been sustained by the frozen ground, which has a low hydraulic permeability. Loss of
permafrost, in turn, promotes drainage (Jepsen et al., 2013). While ponds are already disappearing in some regions, such as
in discontinuous permafrost landscapes in Alaska (Riordan et al., 2006; Andresen and Lougheed, 2015), other regions might
become richer in ponds with warming (Christensen et al., 2004; Bring et al., 2016).

The ice-wedge polygonal tundra is a landscape type that typically features a high pond density. Polygonal tundra covers
roughly 3 % of the landmasses in the Arctic (Minke et al., 2007). It forms because temperatures drop far below freezing in
winter; consequently, the soil contracts and tension cracks open up. These cracks fill with meltwater in the spring before the
soil can expand again. If this process repeatedly occurs, ice wedges eventually form just below the active layer (Jorgenson
et al., 2015). The cracks often occur in shapes that resemble polygons (Cresto Aleina et al., 2013) and the formation of the
ice wedges leads to movement of material from the center of the polygon to the edges resulting in dry rims on top of the ice
wedges and moist centers in the middle of the polygons (Minayeva et al., 2016). Melting of ice wedges is likely accompanied
by increased formation of ponds (Jorgenson et al., 2006; Liljedahl et al., 2016). If the ice wedge itself degrades, a water-filled
trough forms on top. These ponds are often elongated, and the remainder of the ice wedge constitutes part of the pond bottom
leading to cold bottom temperatures. These ponds are labelled ice-wedge ponds. If the middle part of a polygon subsides, in
between the ice wedges, a nearly circular pond develops with a flat bottom. We call these ponds polygonal-center ponds.
Finally, sometimes several polygons subside, leading to comparably large submerged areas, though the polygonal structure
is often visible at the pond edge and bottom. We label these ponds *merged polygonal ponds*. These three pond types exhibit
different methane dynamics (Rehder et al., 2021).

Most Arctic ponds emit predominantly contemporary, recently fixed, carbon (Negandhi et al., 2013; Bouchard et al., 2015;
Dean et al., 2020). However, newly-formed ice-wedge ponds might emit older carbon than the average Arctic pond. When the
permafrost adjacent to the thawing ice wedge degrades, old carbon can leech from the thawed sediments into the pond fueling
methanogenesis (Langer et al., 2015; Prėskienis et al., 2021) and exerting a positive climatic feedback.

Furthermore, the composition of the ponds' methanogenic communities might change in response to the warming Arctic.
Zhu et al. (2020) predicted that this will lead to an additional, strong increase in pond methane emissions. Besides temperature,





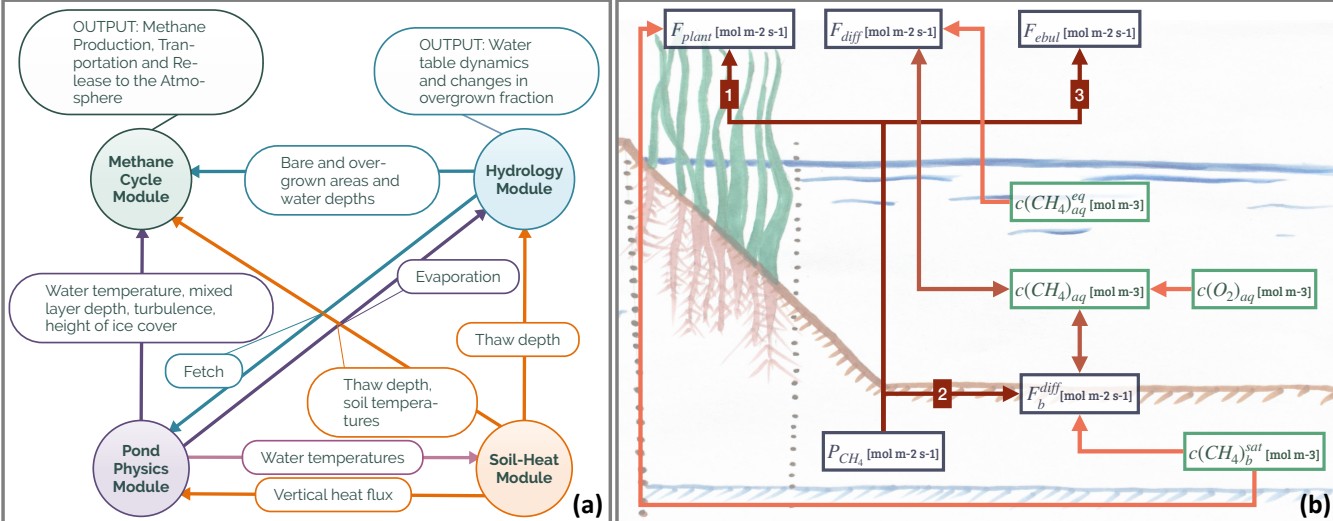

**Figure 1.** (a) Overview of the modules constituting MeEP. The variables which are used to couple the modules are labelling the arrows in between the modules. The output of the methane and hydrological modules, which we use in this work, is listed as well. (b) Overview of the methane module with the main variables. $F_{\text{plant}}$ denotes the plant-mediated transport, $F_{\text{diff}}$ stands for the diffusive flux from water to the atmosphere and $F_{\text{ebul}}$ for ebullition. $P_{CH_4}$ is the rate of methanogenesis, $F_b^{\text{diff}}$ the diffusion from the sediment to atmosphere. Finally, $c$ indicates concentration, the subscript $aq$ labels dissolved gases, $b$ the sediment, and the superscript $eq$ concentration in equilibrium with the atmosphere.

methanogenesis in water bodies depends on substrate quality (de Jong et al., 2018). Vascular plants are known to improve
substrate quality, e.g. by increasing the organic fraction of the soil (Joabsson and Christensen, 2001; Rehder et al., 2021), and
vegetation and its composition in the Arctic are already changing (Villarreal et al., 2012; Bhatt et al., 2013). In consequence
methane emissions from Arctic ponds are expected to undergo substantial changes.

To explore how pond methane emissions might change in a warmer Arctic and analyze as many of these interlinked effects
on methane cycling in a single study as possible, we employ the dedicated model MeEP (Methane Emissions from Ponds). We
focus on the landscape scale (here about $5\ \text{km}^2$) for the polygonal tundra, for which we tuned the model and compared the
output of historical simulations to eddy covariance measurements of pond methane emissions.

## 2    Materials and methods

### 2.1    Short description and set-up of MeEP

MeEP consists of four coupled modules: A pond-physics module, a soil-heat module, a hydrological module, and, the main
focus of this work, a methane module. All modules operate on the same temporal resolution with time steps of one hour,(Fig.
1(a)). The pond physics, hydrological, and methane modules are one-dimensional, while the soil-heat module laterally couples





pond sediments with the surrounding tundra. We set up the model for Samoylov Island in the Lena River Delta, Siberia, and use one instance of the three former modules for each pond type. Each instance of the methane module is split into two parts: One for the overgrown and one for the open-water fraction of the pond. The soil-heat module uses a tiling approach, and we employ

one tile for each pond type and one tile for the surrounding tundra. A detailed description of the methane and hydrological module is included as a supplement to this paper (Fig. S1 and S2). The supplement also contains an overview of the constants (see Tab. S3).

### 2.1.1 Pond physics

We use the lake module FLake (Mironov, 2005) to simulate the physical properties of the pond. FLake is a bulk model

predicting the mixing conditions and the temperature profile of a waterbody. To that end, FLake divides the water column into a mixed layer and a stratified thermocline. FLake incorporates a description of heat transport in the sediment. We switched off this part of the model in favor of the soil-heat module described below, including freeze and thaw processes. Instead, we compute the heat flux from the sediment into the pond based on the equation in FLake by using the temperature profile of our soil-heat module. Thus, the sediment temperatures from the soil heat module provide a heat flux as a lower boundary for

FLake, while water temperature is used as an upper boundary for the soil heat module.

### 2.1.2 Soil heat

We used a simplified version of the CryoGrid permafrost model, called CryoGridLite (Langer et al., 2022), coupled to the FLake model to represent the transient temperature field in the sediments beneath the ponds. Unlike the standard CryoGrid model, this version employs an implicit finite difference scheme to solve the heat equation with phase change, originally

established by Swaminathan and Voller (1992). This allows the representation of a freezing curve for free water with a discrete phase change at $0\,°C$. We emphasize that this is a good first-order approximation for sandy and organic-rich sediments such as those present at the study site. The uncoupled soil-heat model was successfully applied to determine the thickness and shape of taliks beneath serpentine river channels in the Lena-Delta (Juhls et al., 2021). The coupling between FLake and CryoGrid at the top of the sediment domain was achieved by applying the bottom water temperature provided by FLake as the upper

boundary condition to the sediment domain. The lower boundary (at 20m depth) was defined by a constant geothermal heat flux $(0.05\,\mathrm{Wm^{-2}})$. At this depth, local measurements find nearly no annual cycle (Boike et al., 2019). The model framework allows lateral heat exchange with the surrounding permafrost tundra based on laterally coupled tiles (Langer et al., 2016; Nitzbon et al., 2019). We set sediment properties with depth (stratigraphy) individually for the tundra tile and for the pond types. We used local porosity and organic content data from Zubrzycki et al. (2013). Both porosity and organic content decrease with

depths. Under ice-wedge ponds, soil layers starting at 1 m depths consist of 90 % ice.





**Table 1.** Properties of thermokarst ponds on the river terrace of Samoylov Island. Ponds in MeEP are classified as either polygonal-center (PC), ice-wedge (IW) or merged polygonal (MP) ponds. Each of these types is represented by their typical geometry: The average area of an individual pond (mean A), the total area covered by all ponds of a specific type (total A), as well as the overgrown fraction of a pond type (veg. fr.) were provided by the land-cover classification (Beckebanze et al., 2021). The mean depths (mean D) is an estimate by Rehder et al. (2021). $\alpha$ is the angle between the bank of the pond and the horizontal plane. Since macrophytes only grow in shallow water, $\alpha$ was set to match the overgrown fraction of each pond type. Ponds cover roughly 11.5 % of the holocene river terrace of Samoylov Island.

| pond type | mean A [m$^2$] | mean D [m] | $\alpha$ [RAD] | veg. fr. [%] | total A [m$^2$] |
|---|---|---|---|---|---|
| PC | 56 | 0.6 | 0.36 | 53.6 | 136677 |
| IW | 58 | 0.8 | 0.30 | 61.0 | 41172 |
| MP | 1305 | 1.2 | 0.20 | 22.8 | 165819 |

### 2.1.3 Hydrology

The hydrological model (see section S2) is responsible for water-table dynamics fed into FLake and for partitioning of the pond in an overgrown and open-water part for the methane module. Water-table dynamics are computed as the balance between precipitation, evaporation provided by FLake, and above- and below-ground runoff. Below-ground runoff follows Darcy's law, and the soil properties were set according to local hydraulic conductivity measurements by Helbig et al. (2013).

Changes in the water table height lead to changes in the areas of the overgrown and open-water parts of the pond. To compute these changes, we assume the pond's cross-section to be an isosceles trapezoid as a simple geometric form, with an angle $\alpha$ between the slope and the horizontal plane. Plants are assumed to grow in all parts shallow enough (water depths < 0.5 m), and $\alpha$ was set so that the allocation to overgrown and open water matches observations (Tab. 1). The methane module is executed for each part of the pond and uses the respective mean water depth.

### 2.1.4 Methane

The methane module is separated into two parts: One for the ice-free (see Fig. 1(b)) and one for the ice-covered season. In summer, the model is built on three main assumptions:

- We assume equilibrium between production and emission of methane in each time step. Under this assumption, all variables become stationary and time-dependent terms are zero.

- We assume that there is no lateral mixing between the overgrown and the open-water parts of a pond. Thus, we can solve the methane module individually for each part of the pond.

- We assume that the whole water column is well mixed in summer and that the methane concentration throughout the water column is constant.





These assumptions introduce inaccuracies. Due to the first and third assumption, MeEP does not consider any effect of methane

storage in the pond, which we assume to be negligible, because of the small water depths and regular mixing of the water body.

Generally we find that in our simulations, the ponds are completely mixed more than half the time even for ice-wedge ponds,

and that stratification lasts on average less than half a day before the pond is completely mixed again under present conditions.

Under warmer climatic conditions, FLake predicts a further reduction of stratification. Thus, the amount of methane that could

accumulate in the stratified water is limited and stratification most directly impacts the rate of diffusion, only one of the three

pathways for methane. Overall, the assumptions distinctly simplify the model, and we can find an analytical solution to our

equations.

    In MeEP, methane is produced exclusively in the sediment with the production being dependent on the sediment temperature

$T_b$ and thaw depth $h_s$ (Stepanenko et al., 2011) as follows:

$$P_{CH_4} = \frac{P_0}{a} \cdot q_{10}^{(T_b - 273.15)/(10^\circ \mathrm{C})} \cdot (1 - e^{-ah_s}) \cdot f_{\mathrm{prod}} \qquad [\mathrm{mol\ m^{-2}\ s^{-1}}], \tag{1}$$

with $P_0$ being the tuned base productivity of the ponds. The methane production depends linearly on the net primary pro-

ductivity (NPP) through the dimensionless $f_{\mathrm{prod}}$, which is based on Walter et al. (2001) and was set up to range from zero to

roughly one under present day conditions. Since the methanogens do not use all the substrate within the same time step, we

apply a running average on NPP with a window length of one month and split methane production into a part dependent on

NPP (75%) and into a base productivity (25%) based on findings by Bouchard et al. (2015) and Dean et al. (2020). Methane

productivity in the open water correlates with the amount of littoral vegetation (Juutinen et al., 2003), thus for open water

$f_{\mathrm{prod}}$ additionally takes the ratio of overgrown versus open water into account. $a$ [m$^{-1}$] determines how quickly the methane

production decreases with sediment depth, while $q_{10}$ is a constant describing the temperature dependence, which was set to 3.4

according to local measurements (Walz et al., 2017).

All the methane produced in a time step is emitted or oxidised in the same time step through one of three following pathways.

First, in the overgrown part of the pond, evading through emergent macrophytes is the most efficient pathway for methane,

meaning that most methane produced in the sediment is alloted to this plant-mediated transport based on (Walter et al., 1996).

The amount of methane transmitted through vascular plants depends on the thaw depth and leaf-area index as a measure for the

seasonality and density of the vegetation. We assume a fixed fraction of the plant-mediated methane to be oxidized ($f_{\mathrm{ox}} = 0.2$)

reducing the methane flux from plants $F_{\mathrm{plant}}$. The value $0.2$ is an conservative estimate based on the work of Turner et al. (2020)

and Ström et al. (2005), who measured the oxidation rates of the plant species dominating our study region. We compute the

plant-mediated transport as

$$F_{\mathrm{plant}} = (1 - f_{\mathrm{ox}}) \cdot \min\{\beta \cdot f_{\mathrm{growth}} \cdot h_s \cdot c(CH_4)_b^{\mathrm{sat}}, P_{CH_4}\}\ [\mathrm{mol\ m^{-2}\ s^{-1}}]. \tag{2}$$

$\beta$ [s$^{-1}$] is a dimensionless factor describing plant density and their ability to conduct methane combined with an rate factor

(Walter et al., 2001). $f_{\mathrm{growth}}$ is a dimensionless measure of the plant growth which depends on the leaf-area index which varies

between zero and four and is computed following Walter and Heimann (2000). $c(CH_4)_b^{\mathrm{sat}}$ is the saturation concentration of

methane in the sediment, which we compute using temperature-dependent Henry's constants ($H_b^{CH_4}$ and $H_b^{N_2}$). The concen-

tration is controlled by the hydrostatic pressure at the pond bottom $p_h$ and the partial pressure of nitrogen ($N_2$). We assume





the $N_2$ concentration to be in equilibrium with the atmosphere in the water column ($c(N_2)_{eq}$) and decay exponentially in the
sediment with the rate $\lambda_{N_2}$ (Stepanenko et al., 2011; Bazhin, 2001). The saturation concentration then reads

$$c(CH_4)_b^{\text{sat}} = \phi \cdot H_b^{CH_4} \cdot \gamma \cdot \left( p_h - \frac{c(N_2)_{eq}}{H_b^{N_2}} \cdot e^{\frac{-\lambda_{N_2} \cdot h_s}{2}} \right) \qquad [\text{mol m}^{-3}], \qquad (3)$$

where $\phi$ denotes the porosity of the top sediment [$\text{m}^3\ \text{m}^{-3}$], which is set to 0.97 based on measurement data (Helbig et al.,
2013; Zubrzycki et al., 2013) and $\gamma$ a dimensionless threshold, which was tuned using chamber measurements of pond methane
fluxes by Knoblauch et al. (2015).

Next, methane is diffused through the water column and into the atmosphere. We compute diffusion based on the balance

$$F_b^{\text{diff}} - F_{\text{diff}} - F_{ox} = 0, \qquad (4)$$

where $F_b^{\text{diff}}$ and $F_{\text{diff}}$ stand for the methane flux between sediment and water column and between water column and atmosphere,
respectively. Diffusion is the slowest pathway; thus, we dynamically account for oxidation ($F_{ox}$) using the Michaelis-Menthen
relation with constants determined by Martinez-Cruz et al. (2015). We compute $F_b^{\text{diff}}$ based on the gradient between the concen-
tration in water and sediment multiplied with the diffusivity based on Sabrekov et al. (2017). For diffusion to the atmosphere
we utilize

$$F_{\text{diff}} = k_p (c(CH_4)_{\text{aq}} - c(CH_4)_{aq}^{\text{eq}}) \qquad [\text{mol m}^{-2}\ \text{s}^{-1}] \qquad (5)$$

and compute the piston velocity $k_p$ following Heiskanen et al. (2014). $c(CH_4)_{\text{aq}}$ is the water methane concentration and
$c(CH_4)_{aq}^{\text{eq}}$ is the methane concentration if the water column were in equilibrium with the atmosphere. We solve eq. 4 for
$c(CH_4)_{\text{aq}}$ to compute the fluxes.

Lastly, if more methane is produced than what can leave the sediment through plant-mediated transport or diffusion, this
methane escapes the ponds in the form of gas bubbles (ebullition).

We assume that there is no exchange of methane between water column and atmosphere while the pond is ice-capped in
winter. However, methane is still produced in the sediment until it freezes during the ice-on period. This methane accumulates
in the water column, where part of it oxidizes until the oxygen in the water column is depleted. Furthermore, if methane con-
centrations exceed the temperature-dependent saturation concentration, the methane gasses out. This methane is encapsulated
in the ice (Langer et al., 2015). The methane accumulated in the water column and the methane caught in the ice are emitted at
once when the ice cover comes off. For simplicity the ice cover is not fractional, so the pond is either ice covered or has no ice.

## 2.2  Study site in the Lena River delta

In this study we focus on the extensively researched Samoylov Island (Kutzbach et al., 2004; Abnizova et al., 2012; Helbig
et al., 2013; Zubrzycki et al., 2013; Knoblauch et al., 2015; Boike et al., 2019; Rehder et al., 2021; Beckebanze et al., 2021,
among others). Samoylov Island lies in the Lena River Delta of Northeast Siberia at 72°22' N and 126°30' E (Fig. 2). The
island is composed of Holocene sediments and can be divided into two geomorphologically different parts. The western part





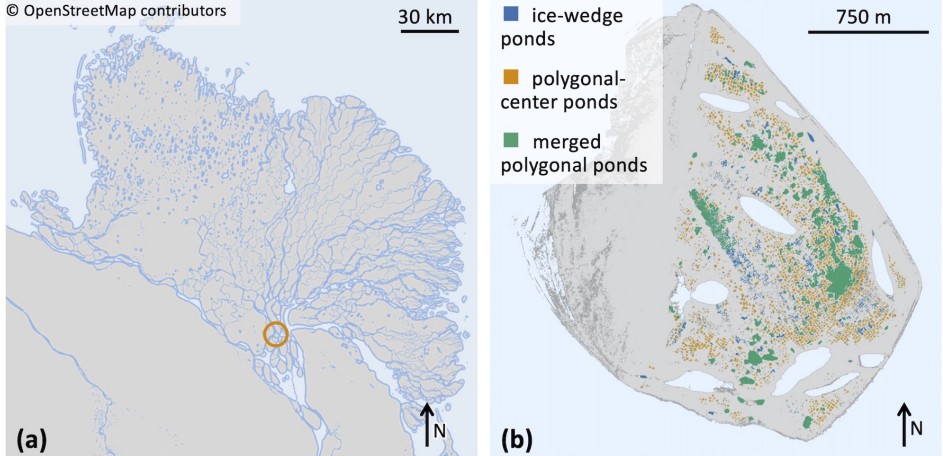

**Figure 2.** (a) Map of the Lena River Delta, which is situated in Northeast Siberia. Location of Samoylov Island marked by a circle. © OpenStreetMap contributors 2022. Distributed under the Open Data Commons Open Database License (ODbL) v1.0. (b) Map of Samoylov Island including a classification of all ponds on the river terrace (eastern part of the island) based on the landscape-scale pond classification. Larger lakes which are not part of this study are drawn in light blue.

consists of a floodplain, while the eastern part is a river terrace featuring polygonal tundra (Zubrzycki et al., 2013; Kartoziia,

2019). This part of the island contains more than 1300 ponds (Muster et al., 2012) in an area of $\sim 3$ km$^2$ (Beckebanze et al., 2021), and thus is an excellent site to study ponds. We use a pond classification (Mirbach et al., 2022) which provides spatial information on the location, size and type of all ponds on Samoylov Island (Tab. 1 and Fig. 2(b)). This classification uses the distinct shapes and sizes of the three pond types to distinguish between the three pond classes. The pond types are determined by size limits and by how compact the shape of the pond is.

**2.3 Forcing and set up of scenario simulations**

To force MeEP, we use a mixture of reanalysis (ERA5, Hersbach et al. (2020)) and remote-sensing (MODIS, Myneni et al. (2015)) data: We use the ERA5 variables for specific humidity, surface downwards solar radiation, surface downwards thermal radiation, surface pressure, temperature at two-meter height, total precipitation and the wind speed at ten-meter height. Wind speed has been computed as the euclidean norm of two orthogonal wind components. From MODIS, we extract the leaf area

index for low vegetation and estimate net primary production. Net primary production is calculated as half of the gross primary production. We always extract the grid box closest to our study site for 2002 - 2019. To spin up MeEP, we compute the average year from this period and force MeEP for ten years with this average forcing. For analysis, we use the years 2004-2019. At the beginning of 2004, the vegetation cover is reset once. In addition to a historical simulation *hist_all*, we simulate warming scenarios. To that end, we scale each forcing variable to fit a $\Delta T$ warmer Arctic, with $\Delta T[°C] \in \{2.5, 5, 7.5\}$ (*exp2.5_all*,

*exp5.0_all*, and *exp7.5_all*). We determine expressions to scale the forcing variables using MPI-ESM simulations (Wieners et al., 2019; Mauritsen et al., 2019) from the 1pctCO2 scenarios of CMIP6 (Eyring et al., 2016). We fit each variable to the





**Table 2.** Overview of warming simulations. The simulations (sim.) we conduct are listed in this table. We use historical forcing for the hist_all simulations, and for the experiments forcing adapted to a mean increase in annual temperature $\Delta T$.

| sim. | hist_all | exp2.5_all | exp5.0_all | exp7.5_all |
|------|----------|------------|------------|------------|
| $\Delta$T [$^\circ$C] | 0 | 2.5 | 5.0 | 7.5 |

**Table 3.** Additional simulations to extract the signal from individual components. We force the methane module with mixed forcing from the hist_all and exp5.0_all simulations, separating three components based on (a) temperature and season length-related variables (*exp5.0_Temp*), (b) variables connected to hydrology (*exp5.0_Hyd*), (c) and variables representing vegetation (*exp5.0_Veg*).

| component | exp5.0_Temp | exp5.0_Hyd | exp5.0_Veg |
|-----------|-------------|------------|------------|
| temp.-related | **exp5.0_all** | hist_all | hist_all |
| hydrology | hist_all | **exp5.0_all** | hist_all |
| vegetation | hist_all | hist_all | **exp5.0_all** |

temperature-related variables: Thaw depth, Deardorff velocity, pond mixed-layer and
bottom temperature, ice thickness and it's changes since the last time step.
hydrology: Areas and mean depths of the overgrown and open-water parts of the ponds.
vegetation: Leaf area index and net primary productivity.

local annual mean temperature for each month, either linearly or with a quadratic function. If the slope of the fit exceeds it's standard deviation, i.e. the change is significant, we scale the forcing variable. No trend was detected for wind speed, shortwave radiation and air pressure. We adapted net-primary productivity, the leaf-area index, incoming longwave radiation,
precipitation, relative humidity and air temperatures.Using the fit, we can compute a monthly change in the forcing for a given temperature increase $\Delta T$. We interpolate linearly between two values to apply this monthly increase to hourly time steps. In the MPI-M 1pctCO2 simulation, our study area warms about 2.1 times faster than the global average. Thus, our local warming scenarios of 2.5, 5, and 7.5 $^\circ$C correspond to moderate global temperature increases of 1.2, 2.4 and 3.6 $^\circ$C compared to present temperatures.

Lastly, to extract the impact of specific components on the pond methane emissions, we simulate the methane module with mixed forcing from the hist_all and exp5.0_all (Tab. 3). The components we extract are (a) temperature and season-length related variables (*exp5.0_Temp*), (b) variables connected to hydrology (*exp5.0_Hyd*), (c) and variables representing vegetation (*exp5.0_Veg*). Since mixing the forcing of the historical simulation and the warming scenario simulations leads to artifacts in spring and fall when ice is about to melt or has just formed, we do not account for the spring flush in these simulations. Thus,
we only focus on open-water season emissions.





**Table 4.** Tuning parameters. The parameters listed below were set using data by Knoblauch et al. (2015).

| Symbol | Value | Unit | Long name |
|---|---|---|---|
| $P_0^v$ | 0.44 | $\mu$mol m$^{-3}$ s$^{-1}$ | Base productivity in vegetated pond fraction |
| $P_0^o$ | 0.11 | $\mu$mol m$^{-3}$ s$^{-1}$ | Base productivity in open-water pond fraction |
| $\gamma$ | 0.26 | - | deviation from ebullition threshold |
| $\epsilon_a$ | 0.046 | m$^3$ m$^{-3}$ | gas-filled porosity in the sediment |

## 3   Model tuning and validation

The base productivity in the sediment and the distribution of methane among the three pathways were tuned using chamber measurements by Knoblauch et al. (2015). Their dataset provides time series of the individual methane pathways for five ponds, four polygonal-center and one ice-wedge pond, on Samoylov Island during two seasons. In total, six variables were tuned (Tab.

4). We tuned the general magnitude of the fluxes using the base productivity $P_0$ (Eq. 1) and tuned it separately for the overgrown and the open-water part of the pond. $\gamma$ is a factor used to determine the saturation concentration in the sediment, which uses Henry's law (Eq. 3). It is introduced as a correction factor to account for the shape of the bubbles; Henry's law was derived for flat surfaces, but bubbles are spherical (Stepanenko et al., 2011). We also tuned the gas-filled porosity in the sediment, for which no measurements were available. This parameter influences the diffusion from the sediment into the water column (Sabrekov et al., 2017). When comparing the individual flux measurements against modeled values (Fig. 3), we achieved an R$^2$

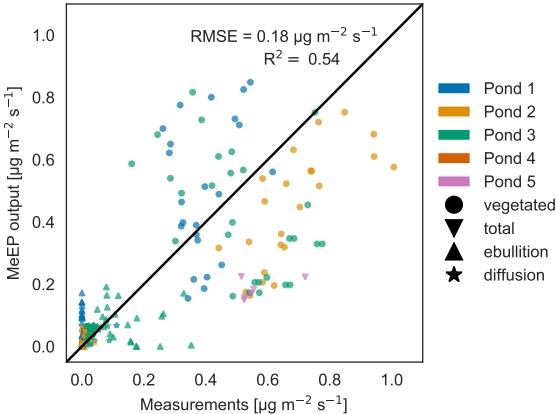

**Figure 3.** Measured versus tuned modeled methane emissions. Comparison of measured (x-axis) and modeled (y-axis) methane fluxes for the five ponds measured by Knoblauch et al. (2015) (color code). The fluxes are broken down into different pathways (ebullition and diffusion) where possible. Vegetated fluxes are fluxes measured over the overgrown part of the pond.


value of 0.54, and thus is able to capture the average behaviour of the ponds. Notably, the maximal goodness of the fit we can





achieve depends on the $q_{10}$ value we use. We set $q_{10}$ to 3.4 to match local measurements (Walz et al., 2017). Jansen et al. (2022) synthesized measurements of methane production in global lake sediments and determined the temperature dependency using an Arrhenius-type equation. In the temperature range of our simulations, using a $q_{10}$ of 3.4 is in the range of the uncertainty of

Jansen et al. (2022) (see Fig. S3). However, if we use a $q_{10}$ of 2 and tune the model again, we achieve a better match between model and measurements ($R^2$ value of 0.63, see Fig. S4). This indicates that either the temperature dependence of methane production is lower than 3.4 or that we underestimate the temperature dependency of methane consuming processes along the different emission pathways. However, both tuned model versions, with an $q_{10}$ of 2.0 and of 3.4 have a similar annual cycle (Fig. S5(a)), because we tune the magnitude of summer emissions to measurements. Nevertheless, the spread between total

annual emissions is larger when using a higher $q_{10}$ (Fig. S5(b) and S6) and the standard deviation in the estimate by Walz et al. (2017) can lead to a doubling or halving in total emissions, if the model is not tuned again (Fig. S6). This shows that the measurement uncertainty regarding the temperature dependence of methane production translates into large uncertainty regarding modeled methane emissions if the modeled emissions are not constrained by further measurements. To assess how

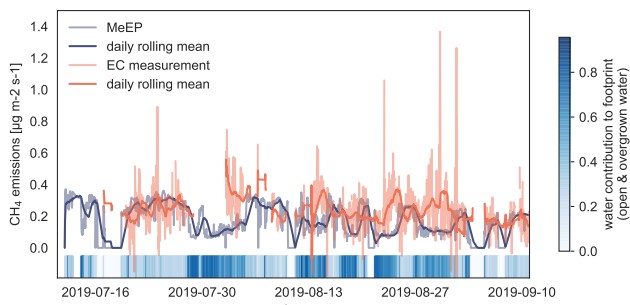

**Figure 4.** Validation of MeEP with eddy covariance (EC) measurements. We compare the EC flux measurements (Beckebanze et al., 2021) to simulated EC flux using the overgrown and open-water fluxes modeled with MeEP and the measured mean tundra fluxes of $0.15\ \mu g\ m^{-2}$ $s^{-1}$ multiplied with their respective contribution to the footprint. The eddy covariance measurements were taken next to a large merged polygonal pond. To visualize how much the pond contributed to the flux in a time step, we added colored strips at the bottom of the plot.

well the model performs compared to measurements to which it has been tuned, we use eddy covariance measurements from

Samoylov Island from the summer 2019 (Beckebanze et al., 2021). Eddy covariance fluxes are almost always a compound of fluxes from different landcover classes. In this case, the footprint, the area measured by the eddy covariance instruments, includes mostly tundra to the west interspersed with polygonal-center ponds, which make up about 10 % of the footprint in this wind direction. To the east, the footprint of the tower consists of the open and overgrown water of the merged polygonal pond to over 90 %. The relative contribution of the three surface classes, open and overgrown water and tundra, to the eddy

covariance flux varies with time and a pure signal from the water body does not exist. To compare MeEP to measurements, we imitate the eddy covariance signal using the contribution of each of the three surface classes to the footprint. This contribution was retrieved using a footprint model and a landcover classification (Mirbach et al., 2022). The overgrown and open-water fluxes predicted by MeEP are multiplied with their respective cover fraction. To this, we add the mean tundra flux determined



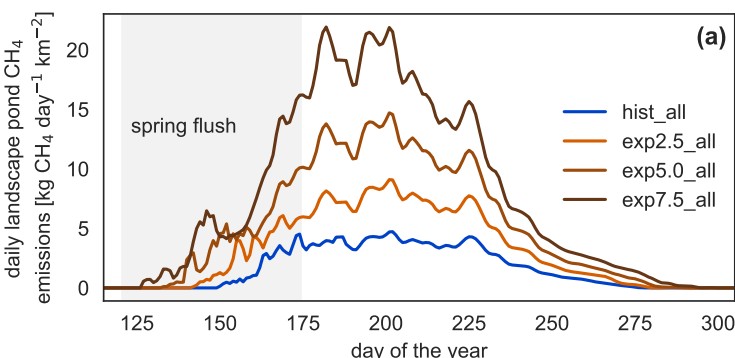 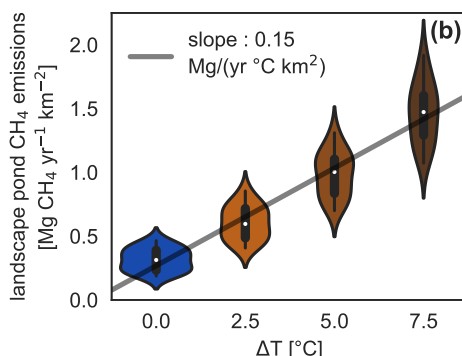

**Figure 5.** Daily and annual methane emission changes in the different simulations. (A) Seasonal dynamics of methane emissions from ponds per square kilometer of river terrace on Samoylov Island in scenario simulations. The seasonal cycle exhibits a peak at the beginning of the open water season caused by the spring flush. (B) Linear regression of annual landscape-scale pond methane emissions per square kilometer river terrace versus annual mean temperature increase. The distribution of annual emissions per year in each simulation is depicted as violin plots: The more often a certain y-value occurs, the wider the shape is.

with the eddy covariance method of $0.15$ $\mu$g m$^{-2}$ s$^{-1}$ and then compare this simulated eddy covariance flux to the real eddy

covariance fluxes (Fig. 4) Please refer to Beckebanze et al. (2021) for more detail on the data processing.

MeEP-based fluxes are slightly lower, so MeEP output might be a conservative estimate of landscape pond methane emissions. However, there are some differences in temporal development. The spatial heterogeneity likely causes these differences in the measured fluxes, which MeEP can not reproduce. Seep-ebullition (constant ebullition from one spot) likely generated especially high emissions from one point in the measurements. In the simulated fluxes, ebullition is assumed to be constant over

the area. Thus, differences in the temporal evolution are expected, and we conclude that the tuning of MeEP was successful.

We want to note that MeEP was designed for an average pond, not for individual ponds. Methane emissions from individual water bodies can be highly variable (Sepulveda-Jauregui et al., 2015; Jansen et al., 2020; Beckebanze et al., 2021). However, MeEP provides emission estimates for an average pond rather than resolving spatial heterogeneity within a pond.

## 4 Results

### 4.1 Methane emission response to warming

MeEP projects an increase of methane emissions with warming (Fig. 5), from methane emissions of $(316\pm86)$ kg CH$_4$ year$^{-1}$ km$^{-2}$ (mean and standard deviation) in the hist_all simulation, to $(605\pm128)$, $(985\pm172)$ and $(1466\pm232)$ kg CH$_4$ year$^{-1}$ km$^{-2}$ in the exp2.5_all, exp5.0_all and exp7.5_all simulations, respectively. These are the average river terrace emissions, using an area-weighted mean of the three pond types. Emissions in the exp5.0_all simulation are 3.1 times higher than in the



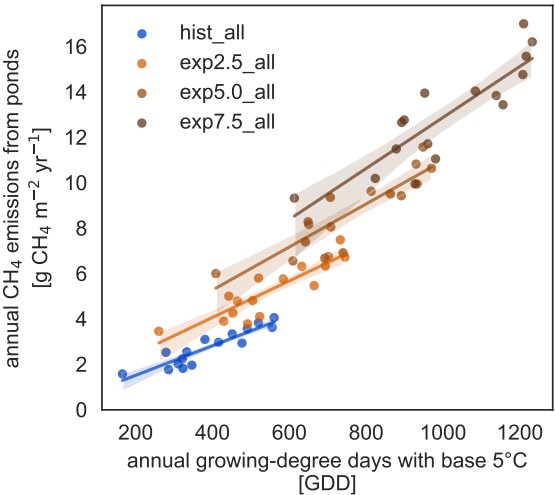

**Figure 6.** Growing-degree days as a control on annual methane emissions. For each simulation, the dependence of the cumulative annual methane emissions (y-axis) on the cumulative annual GDD5 (x-axis) can be approximated by a linear regression (solid lines, confidence intervals shown as shaded area).

hist_all simulation. Using a linear regression between the mean increase in annual air temperature and the total pond emissions (Fig. 5(B)), we determine an increase of emissions from pond areas of $1.33$ g $CH_4$ year$^{-1}$ $^{\circ}C^{-1}$ m$^{-2}$. The increase in annual emissions is caused by an increase in mean emissions over the open-water season and by a longer open-water season (Fig. 5(A)). The open-water season lengthens from a mean of 109 days in the hist_all, to 124 days in the exp2.5_all, 138 days in the exp5.0_all, and 152 days in the exp7.5_all simulation. On average, the growing season lengthens by 5.7 days per degree

of warming. We further investigate the interaction of increased season length and elevated temperatures using growing-degree days. The accumulated growing-degree days above $5\,^{\circ}C$ (GDD5) integrate temperatures and season length in one metric. The annual methane emissions exhibit a clear linear dependence on GDD5 (Fig. 6). This linear dependence, however, does not hold for all simulations. The differences between the applied forcings cause offsets between the different experiments. While the forcing itself uses ERA5 and MODIS, we used ESM scenarios simulations to determine the forcing variable's sensitivity to

warming. We find a strong dependence, especially of net primary productivity on warming in the ESM simulations, leading to pronounced changes in this variable across MeEP simulations. If two years, one from the hist_all simulation and one from exp2.5_all, have a similar mean annual air temperature, then net primary productivity in the exp2.5_all simulation will exceed primary productivity in the hist_all simulation by more than 60%. Thus, air temperature can only be used as a proxy for other forcing variables within one simulation, not across simulations with different forcing. However, air temperature or GDD5 can

predict total pond methane emissions within one simulation. Note that, in contrast to Fig. 5, the emissions displayed in Fig. 6 are not integrated over the total pond area on the Samoylov Island but are given in relative units per pond area. Thus, we do not account for changes in the pond area.





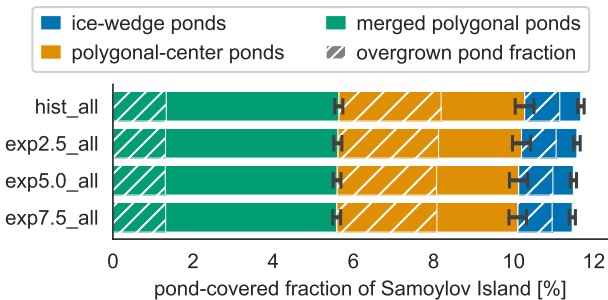

**Figure 7.** Pond area changes between simulations. The average landscape fraction covered by each pond type (y-axis) changes slightly between scenarios (x-axis). The overgrown fraction of each pond type is hatched.

## 4.2 Hydrological response to warming

The total pond area and its allocation to open and overgrown water change with time between simulations. In MeEP, ponds are
initialized at the beginning of the simulation. Though no new ponds can form during a simulation, MeEP computes the water table based on the hydrological budget of precipitation, evaporation, and both below- and above-ground runoff. MeEP projects only a small reduction in the total pond area in response to elevated temperatures (Fig. 7). Even the area reduction of the most extreme warming simulation we conducted (exp7.5_all) is still within the standard deviation of the base simulation hist_all. Total pond areas decrease from a landscape fraction of $11.7 \pm 0.4$ (mean $\pm$ standard deviation) % in the hist_all scenario to
$11.5 \pm 0.4$ % in exp7.5_all. Consequently, the changes in the areas of open and overgrown water are negligible, and on average, $4.7 \pm 0.4$ % of the landscape is covered by the overgrown water fraction of ponds. However, the hydrological module of MeEP is rather simple, and we will examine its limitations in the discussion section 5.3.

## 4.3 What causes the methane emission to increase?

Since the changes in waterbody areas are small, the impact of the hydrology on the total methane emissions is small too.
Nevertheless, the decrease in area is the only response of the system, which leads to a reduction of the emissions under warming (Fig. 8). Changes in physical variables, like temperature and the length of the open-water season, lead to an increase in emissions with warming. In our simulation, vegetation biomass and productivity increase under elevated temperatures. These changes in vegetation are the dominant driver of increased methane emissions with warming. Though emissions start later in the year in the exp5.0_Veg simulation, the total annual emissions are much higher than in the simulations which exclude
the increased plant productivity. Thus, changes in the forcing related to vegetation are the main driver of methane emission increases.

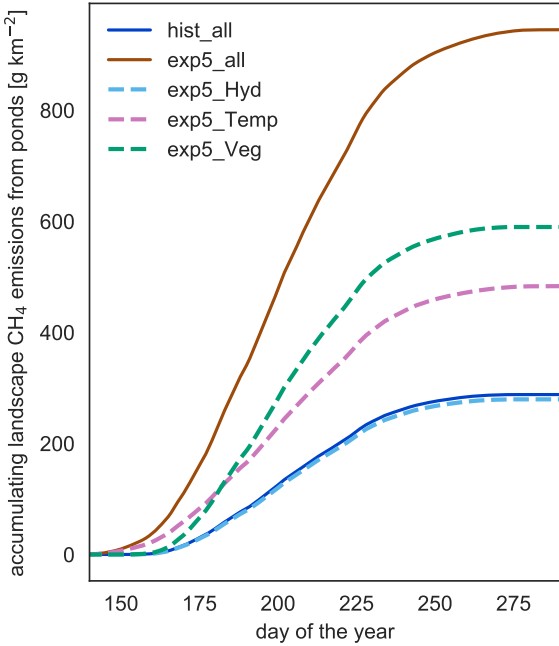

**Figure 8.** Components driving changes in methane emissions. To isolate the impact of different component on the increasing methane emissions, we simulate the methane part of MeEP mixing input from hist_all and exp5.0_all (see Tab. 3). In this figure, we compare the accumulated emissions over the course of the average year between different simulations.

**Table 5.** Methane emissions from each pond type. The open-water season fluxes differ between open and overgrown water for the three pond types. Shown here are values from the hist_all simulation. The total pond fluxes are computed using an area-weighted mean.

| | $CH_4$ fluxes [mg day$^{-1}$ m$^{-2}$] | | | | | | | | |
|---|---|---|---|---|---|---|---|---|---|
| | **total pond** | | | **overgrown fr.** | | | **open-water fr.** | | |
| | **min** | **med** | **max** | **min** | **med** | **max** | **min** | **med** | **max** |
| **IW** | 4.93 | 26.80 | 187.42 | 7.04 | 38.09 | 251.73 | 0.0039 | 8.09 | 74.16 |
| **PC** | 4.54 | 25.28 | 222.73 | 7.03 | 39.60 | 336.72 | 0.0050 | 8.16 | 88.68 |
| **MP** | 2.10 | 12.21 | 76.37 | 8.15 | 40.73 | 222.55 | 0.0008 | 3.68 | 34.74 |

IW, ice-wedge pond; PC, polygonal-center pond; MP, merged polygonal pond.

min, minimum; med, median; max, maximum; fr, fraction.

## 4.4 Impact of pond methane emissions on the landscape scale

The impact of vegetation can also be observed when investigating the impact of overgrown- and open-water fluxes on the landscape scale. While the modeled fluxes from overgrown water exceed the measured average tundra fluxes (Wille et al.,
2008), fluxes from open water are lower (Fig. 9 (A)). When comparing simulated emissions from the three pond types to

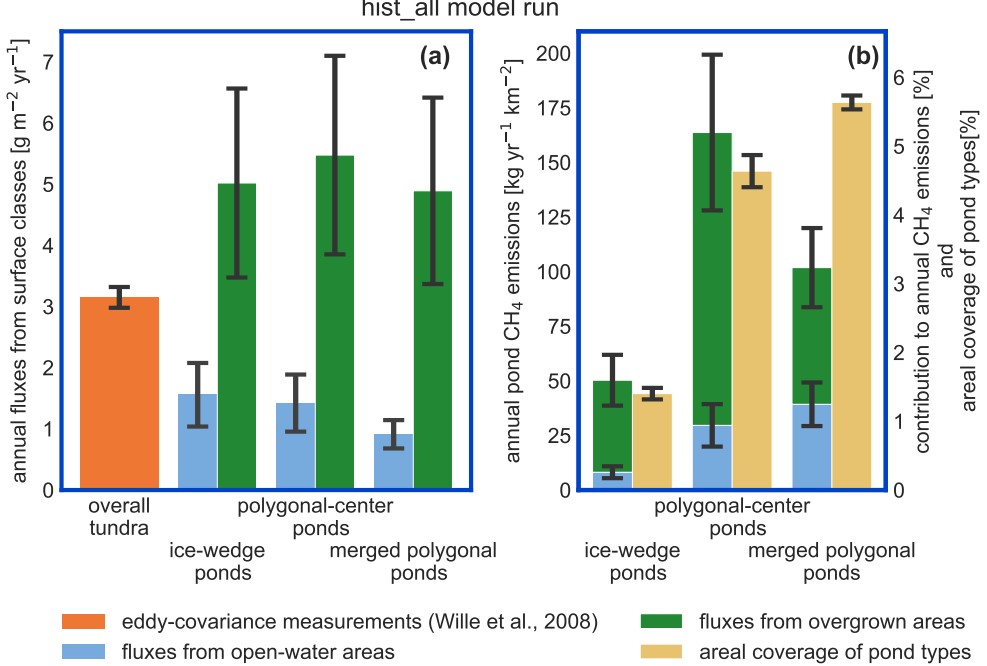

**Figure 9.** Impact of pond emissions on landscape methane emissions. (A) For the hist_all simulation, we compare fluxes per area from different landscape elements. The estimate for the overall tundra emissions were acquired with eddy-covariance measurements over the growing season of 2003 (Wille et al., 2008). Note, that the influence of ponds on these measurements is low. The methane emissions per square meter of open and overgrown water are broken down per pond type. (B) Methane emissions per square kilometer of river terrace of each pond type are displayed as stacked bars. We compare these emissions per pond type to the area this pond type covers on the river terrace of Samoylov Island (sand-colored bar). This comparison relies on the assumption that the emissions measured by Wille et al. (2008) are representative for river-terrace emissions.

overall tundra emissions measured with eddy covariance (Fig. 9 (B)), we find that ice-wedge and polygonal-center ponds emit slightly more methane per unit area of pond than the average tundra. In contrast, merged polygonal ponds emit slightly less. The latter are the pond type with the highest open-water fraction (Tab. 5). To summarize, though small ponds contribute slightly out-of-proportion to the landscape methane emissions, we do not find that ponds are hot spots of methane emissions in the

landscape scale - at least not under the current climate.

Albeit lower than the fluxes from overgrown water in all scenarios, open-water fluxes become more important in the warming simulations (Fig. 10), mostly due to increased ebullition. The relative importance of the plant-mediated fluxes, on the other hand, stays constant over the scenarios, while the impact of the spring flush decreases substantially. In hist_all, the spring flush contributes between 6 - 23 % (minimum and maximum) with a mean contribution of 10 %. In the exp7.5_all simulation, the





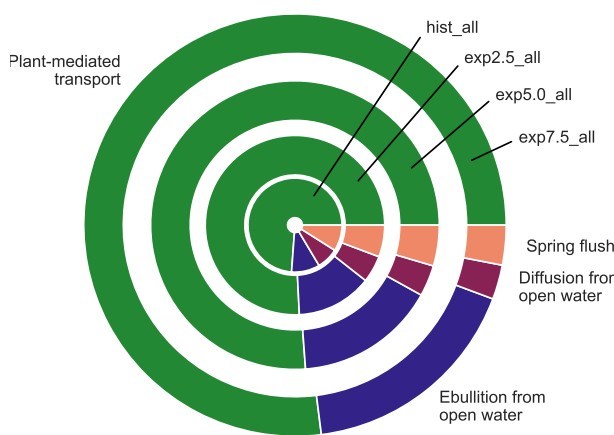

**Figure 10.** The contribution of each flux type to the overall emissions in the average year. The size of a segment represents the contribution of the respective flux type. The area of each circle is proportional to the absolute methane emissions of the average year in each simulation.

maximal contribution of the spring flush of 5 % is lower than the minimum contribution in the hist_all simulation. Overall, we find a pronounced increase in pond methane emissions with warming.

## 5 Discussion

### 5.1 MeEP model output is conservative compared to pan-Arctic methane flux measurements

Ponds in our study region exhibit low methane emissions compared to other Arctic ponds (Liebner et al., 2011). Emissions from ponds are of a similar magnitude as the overall tundra emissions measured by an eddy covariance system which averages over different surface types (Fig. 9, Beckebanze et al. (2021)). MeEP was tuned to local measurements reproducing the typical local emissions patterns and predicts median emissions of 12 - 27 mg day$^{-1}$ m$^{-2}$ depending on the pond type. In a circumpolar synthesis, Kuhn et al. (2021) determined methane emissions from lakes of classes. They estimated median diffusive fluxes of 16 mg day$^{-1}$ m$^{-2}$ from small ($< 0.1$ km$^2$) peatland lakes, the class closest to the water bodies studied here. Adding ebullitive fluxes of 23 mg day$^{-1}$ m$^{-2}$, they estimate total emissions of 39 mg day$^{-1}$ m$^{-2}$ from small peatland lakes. Thus the present-day emissions in our study area are low compared to pan-Arctic average and when upscaling to a broader landscape, higher emissions than the ones predicted here can be expected.

The emission patterns of pond types in MeEP are the same as in observational studies. Bouchard et al. (2015) found open-water emissions with a median of 56.6 mg day$^{-1}$ m$^{-2}$ from ice-wedge ponds, 27.9 mg day$^{-1}$ m$^{-2}$ from polygonal-center ponds and 3 mg day$^{-1}$ m$^{-2}$ from lakes in a polygonal landscape in northeastern Canada in July. As in our model setup,





ice-wedge ponds emit only slightly more methane than polygonal-center ponds, and larger water bodies emit considerably less.

Prėskienis et al. (2021) also measured the spring flush. They estimate that up to 52 % of the annual methane is emitted when the ice melts. This is nearly double of our maximum values of 23 %. Wik et al. (2016), who aggregated pan-Arctic fluxes report an average spring flush of 27 % for thermokarst water bodies, which is still larger than the mean amount in the hist_all simulation of 10 %. Notably, Wik et al. (2016) summarized water bodies of all sizes in the thermokarst category, and Prėskienis et al. (2021) reported lower spring flushes for larger water bodies. Thus, the spring flush modeled by MeEP (Fig. 5) might be a low estimate of the real spring flush but are in the right order of magnitude.

Pond methane emissions from our study site are lower than emissions measured elsewhere and the spring flush predicted by MeEP is also a lower estimate. So, while pond methane emissions in the hist_all simulation are comparable to measurements in, e.g., the Canadian Arctic, the absolute magnitude of fluxes presented in this study are a conservative estimate compared to pond emissions on the pan-Arctic scale.

## 5.2 Vegetation changes intensify pond methane emission increases

In the whole Arctic, vegetation has a strong impact on methane emissions (Joabsson et al., 1999; Andresen et al., 2017; Turner et al., 2020, e.g.). We can split this impact into two parts. First, vascular-plant productivity increases substrate availability, which increases methanogenesis (Joabsson and Christensen, 2001; Kim, 2015). Second, emergent macrophytes are a highly efficient pathway for methane emissions (Knoblauch et al., 2015).

Plant-mediated transport means gases diffuse with low resistance through the aerenchyma of plants. Aerenchyma are air-filled pores in leaves, roots, and stems of macrophytes (Whiting and Chanton, 1992; Colmer, 2003). The methane flux through these plants increases with their above-ground biomass (Ström et al., 2012; Joabsson and Christensen, 2001), and the above-ground biomass correlates linearly with leaf-area index (Andresen et al., 2017). MeEP uses leaf-area index, a variable readily available from remote sensing (Myneni et al., 2015), to modulate the plant-mediated transport (Walter and Heimann, 2000). The leaf-area index increases with temperature in our forcing (Euskirchen et al., 2009). This trend is in line with findings that emergent macrophytes, for example *Arctophila fulva*, already have become more abundant in some regions. *Arctophila fulva* is a very efficient transmitter of methane (Knoblauch et al., 2015; Andresen et al., 2017), which is also abundant in our study region (Knoblauch et al., 2015). This emergent macrophyte is already expanding in similar landscapes, e.g., on Barrow peninsula in Alaska (Villarreal et al., 2012). We find that coverage of emergent macrophytes increases in such a way that plant-mediated transport is limited by methanogenesis rather than by the conductivity and abundance of aerenchyma. In overgrown parts of the ponds, plant-mediated transport is by far the dominant mode of transportation (Whiting and Chanton, 1992; Andresen et al., 2017). When comparing open-water and overgrown fluxes, the contribution of the overgrown part stays constant over all scenarios with increasing methane emissions (Fig. 10). The plant-mediated transport scales with the increase in total emissions because the density of vascular plants increases with temperature. We represent the density of plants using leaf-area index in the model. A higher density of vascular plants means a higher density of aerenchyma, which increases the capacity of the plant-mediated transport more efficiently. Thus, this capacity builds up at the same rate as methanogenesis.




Further, vegetation in permafrost regions adds a positive feedback loop to warming (Lara et al., 2019). Higher temperatures increase plant biomass in the Arctic (Euskirchen et al., 2009; Elmendorf et al., 2012; Andresen and Lougheed, 2015), and, with an increasing thaw depth, conditions for plants become more favorable: Nutrients, which are a limiting factor in tundra landscapes, leach out of the thawing permafrost and support vegetation growths (Andresen et al., 2017; Lara et al., 2019). A higher macrophyte cover adds more substrate to the sediment fueling methanogenesis (Joabsson et al., 1999; Joabsson

and Christensen, 2001; Ström et al., 2012; dos Santos Fonseca et al., 2017), which already under present conditions consumes mostly contemporary carbon (Negandhi et al., 2013; Dean et al., 2020). This increases methane emissions, closing the feedback loop (Lara et al., 2019).

   In MeEP, this increase in substrate is the main driver of elevated emissions under warming (Fig. 8), leading to an increase in methane emissions that is more than 20 % higher than the increase due to higher temperatures alone. The strength of the

methanogenesis response to warming is determined by the term $f_{\mathrm{prod}}$ (see Eq. 1). This term prescribes a linear dependence of methane production on relative changes in net primary productivity. A connection between plant productivity and methanogenesis has been observed in a subarctic fen (Whiting and Chanton, 1992). However, this connection is species-dependent (Vizza et al., 2017; Ström et al., 2005), and some species typical for European wetlands can also reduce methanogenesis (Grünfeld and Brix, 1999). To improve the model, we need additional studies on the impact of emergent macrophytes on Arctic pond or

lake methanogenesis.

   The linear dependence of methanogenesis on plant productivity is a reasonable first estimate given the evidence that in Arctic landscapes, vascular plants enhance methanogenesis (Joabsson and Christensen, 2001; Ström et al., 2003; Lara et al., 2019). A parameterization based on new measurements that focus on macrophytes' impact in ponds on methanogenesis would be a step forward to constrain future pond methane emissions better. A dynamic model of macrophyte coverage and productivity

could be included in a second step. Despite uncertainty in the strengths of the link between methanogenesis and vascular-plant productivity, our projections underpin the importance of future vegetation changes for pond methane emissions.

   Vegetation changes occur slowly on multi-annual timescales (Villarreal et al., 2012), leading to higher emissions even in comparably cool years. This effect is especially apparent in Fig. 6: The regressions do not collapse onto a single line. Rather years with the same growing-degree days emit more methane with higher warming. However, growing-degree days are a good

predictor of annual methane emissions within a simulation. They combine the direct impact of temperature with a measure of how favorable temperatures are for plant growth for each year. In the Arctic, multi-year vegetation changes are already well underway (Bhatt et al., 2013; Wrona et al., 2016). However, vegetation changes in the Arctic do not solely depend on temperature (Wrona et al., 2016), and the Arctic does not become greener in all regions, but also browns in some (Bhatt et al., 2013; Winkler et al., 2021). This browning is strongly connected to changes in hydrology as browning is caused by a lack of

water(Winkler et al., 2021).

### 5.3    Hydrological changes slightly decrease pond methane emission

The tendency of a landscape to either become wetter or drier under warming is dependent on local topography (Jones et al., 2022; Miner et al., 2022). An overall inclined area is likely to drain (Bring et al., 2016). However, in a very flat landscape





such as our study area, it might get wetter with warming (Christensen et al., 2004). In the polygonal tundra, warming leads
to permafrost degradation, which prompts loss of ground ice, subsidence, and pond formation leading to higher methane
emissions (Kim, 2015), especially along the ice wedges (Yoshikawa and Hinzman, 2003; Liljedahl et al., 2016). Consequently,
a degrading polygonal tundra features an increasing number of ice-wedge ponds (Bouchard et al., 2020; Wickland et al., 2020).
As the degradation proceeds, the ponds are inclined to vanish again, either because of infilling or drainage (Stow et al., 2004;
Cresto Aleina et al., 2015; Jorgenson et al., 2015; Wickland et al., 2020). Additionally, an increase in emergent macrophytes
can promote pond drainage by intensifying transpiration (Andresen and Lougheed, 2015).

The landscapes drain as permafrost thaws because the disappearing ice has been acting as in barrier for the water. Without
permafrost, water can better drain subsurface. However, drainage is impeded if the soils have low permeability, such as highly
decomposed peat, and ponds and lakes can be sustained (Smith et al., 2005). Though we do not focus on pond formation in
MeEP, existing ponds may drain. MeEP includes a simple surface and subsurface flow formulation, which depends on the local
permeability (Helbig et al., 2013). In MeEP, pond areas decrease slightly with warming (Fig. 7). Thus, even in our first-order
approximation of pond hydrology, we find evidence of pond drainage reducing pond methane emissions (Fig. 8), though to a
lesser extent than for example van Huissteden et al. (2011). They reported that drainage limits water body methane emissions
on the landscape scale. The hydrological model implemented in MeEP is one-dimensional and can consequently only provide
a first-order estimate of water-table dynamics. More complex dynamics on the landscape scale, such as the formation of a
network of channels along the ice wedges promoting fast drainage through percolation (Cresto Aleina et al., 2013). Thus, our
estimate of runoff might be too low.

### 5.4   Landscape-scale impact of pond methane emissions

When estimating the landscape-scale impact of methane emissions from ponds and lakes, many studies concentrate on diffusive
emissions (Juutinen et al., 2009; Holgerson and Raymond, 2016; Polishchuk et al., 2018; Hughes-Allen et al., 2021; Zabelina
et al., 2020), though some also include ebullition (Sepulveda-Jauregui et al., 2015; Wik et al., 2016; Kuhn et al., 2021). We
find that including ebullition is important because, in ponds, ebullition contributes more than diffusion to the total emissions
(Kuhn et al., 2021; Praetzel et al., 2021) and becomes much more important with warming (Fig. 10). In MeEP, ponds are very
sensitive to rising temperatures. The model projects emissions to roughly double at a temperature increase of only 2.5 °C (Fig.
5(B)).

Much of the intensification of methane emissions in MeEP is due to vegetation growth, which leads to a strong boost in mean
emissions during the ice-free season and a higher peak of emissions in summer (Fig. 5(A)). These emissions are already under
current climatic conditions notably higher than mean tundra emissions (Fig. 9) and should be included in future large-scale
ponds methane emissions assessments.

We might even underestimate the response of ponds to warming because methane production is described by a q$^{10}$-equation
Walz et al. (2017). This description does not account for shifts in methanogen communities, which can enhance the rate of
methanogenesis under warming (Zhu et al., 2020). Additionally, we only account for present-day substrate in the current setup:
Methanogenesis is coupled to vegetation productivity of the same year. This assumption is valid for ponds at the moment

(Negandhi et al., 2013; Bouchard et al., 2015; Dean et al., 2020), but might change as permafrost degrades and old carbon leeches into the ponds (Langer et al., 2015; Prėskienis et al., 2021). This additional carbon is not included in our projections.
Therefore, our estimate is conservative.

## 6 Conclusions

While ponds are not hotspots of methane emissions in our study area under the current climate, our model simulations indicate that they will become stronger methane sources under further warming. We project an increase of pond methane emissions of 1.33 g $CH_4$ m$^{-2}$ year$^{-1}$ °C$^{-1}$. At the same time, the pond area decreases only slightly. However, the hydrological module
of MeEP only gives a first-order approximation of water-table dynamics. To better gauge the future impact of ponds, we need better projections of pond inception and drainage.

Much of the methane-emission increase from ponds is mediated through macrophytes. The vascular plants become more productive and provide additional substrate for methanogenesis. In our simulations, the impact of the additional substrate on methanogenesis is substantially stronger than the impact of elevated temperatures or a prolonged open-water season. However,
the relationship between emergent-macrophyte productivity and methanogenesis in ponds could only be approximated due to a lack of measurement data. We further find that plant-mediated transport is the methane pathway contributing most to the overall landscape emissions in simulated temperature regimes. Unfortunately, plant-mediated transport is the methane pathway least often reported in measurement datasets of pond methane emissions. This makes it harder to generalize our findings to a larger scale, and more observations of this emissions pathway and its contribution to overall pond methane emissions are
needed. Additionally, the current version of MeEP only uses one value for the conductivity of plants, even though we know that different plant species conduct methane with varying efficiency (Knoblauch et al., 2015). To upscale the plant-mediated fluxes realistically, vegetation maps of the dominant macrophytes would be a strong asset. However, we suppose that vegetation similarly impacts ponds in other Arctic regions. In that case, it is crucial to include macrophytes as a substrate source and as an efficient methane pathway for a pan-Arctic assessment of pond methane emissions under warming.

*Code and data availability.* During the review process, primary data, scripts and model code can be obtained from the author. Upon acceptance of the paper, primary data and scripts used in this study will be archived by the Max Planck Institute for Meteorology and can be obtained by contacting publications@mpimet.mpg.de. Model code will be published under an open access license with an DOI through zenodo.

*Author contributions.* ZR: conceptualization, methodology, software, formal analysis, investigation, writing - original draft, review & edit-
ing, visualisation. TK: conceptualization, formal analysis, writing - review & editing, supervision. LK: formal analysis, writing - review & editing, supervision. VS: conceptualization, methodology. ML: methodology, software, writing - original draft. VB: conceptualization, formal analysis, writing - review & editing, supervision.



*Competing interests.* The authors declare no conflicts of interest.

*Acknowledgements.* Thanks to ICDC, CEN, University of Hamburg for data support. This work was funded by the German Research Foundation as part of the CLICCS Clusters of Excellence (DFG EXC 2037). This work contributes to the European Research Council (ERC) under the European Union's Horizon 2020 research and innovation program (grant agreement No 951288, Q-Arctic). TK acknowledges support from the German Federal Ministry of Education and Research (BMBF), Research for Sustainability initiative FONA, through the project PalMod (Grant No. 01LP1921A).



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
