# Peer review of "Simulated methane emissions from Arctic ponds are highly sensitive to warming"

_Biogeosciences, 2022_

## Author Comment (AC1)

**Review #1**

Dear reviewer,

Thank you very much for your positive and constructive feedback and comments on our manuscript. We tried to incorporate your suggestions into the paper. Please find below your comments in orange and our point-by-point response in black. Updated excerpts of the manuscript are included in blue.

The manuscript by Rehder et al. Employs a process-based model for methane emissions from ponds to investigate who methane emissions from polygonal-tundra ponds will be affected by warming. This is a very important topic, especially now that we know that the Arctic has been warming faster than any other region in the world and that this warming may let to the proliferation of thaw ponds. Although the model seems to be robust, my major concern is the applicability of the model to other sites to be able to extrapolate the findings to other regions. Moreover, what about the representativity of the selected ponds regarding ponds in the Arctic? In my opinion, a new section discussing how to upscale the results would be useful. Also, a sensitivity analyses of the models needs to be added.

> Thank you for your kind summary. Please refer to the specific comments regarding the upscaling and the sensitivity analysis. Reviewer #2 also asked about upscaling so there is additional information in our response to that review as well. We added an outlook on how to use the model on larger scales.

**Specific comments:**

Abstract

Line 1  I suggest starting the abstract with the broad picture, i.e., mentioning that ponds are important sources of CH4 and vulnerable to warming. After that, authots can focus on the objective of your study.

> We rearranged the first three sentences of the abstract in the following way:
>
> The Arctic is warming at an above-average rate, and small, shallow water bodies such as ponds are vulnerable to this warming due to their low thermal inertia compared to larger lakes. While ponds are a relevant landscape-scale source of methane under the current climate, the response of pond methane emissions to warming is uncertain. We employ a new, process-based model for methane emissions from ponds (MeEP) to investigate the methane-emission response of polygonal-tundra ponds in Northeast Siberia to warming.

Line 5  I think the abstract contain too many details about the model. That should be placed in the Methods section. Focus on the main characteristics of the model, its novelty, and the main results.

> We shortened the part about the model description in the following way:

MeEP is the first dedicated model of pond methane emissions which differentiates between the three main pond types of the polygonal tundra and resolves the main pathways of methane emissions – diffusion, ebullition, and plant-mediated transport.

Line 9 Why these warming temperatures were chosen? Explain this later in the methods.

The range was chosen to reflect likely future temperatures. We added the following sentence in the methods:

With this evenly spaced temperature increases, we track how emissions change within a maximum warming the Arctic would be exposed to even under moderate global warming due to Arctic amplification (Bekryaev et al., 2010, Rantanen et al., 2022).

Line 10 Mention here that this increase in warming seems to be linear with temperature.

We added this information to the sentence.

The simulations reveal an approximately linear increase of emissions from ponds of 1.33 g $CH_4$ year$^{-1}$ °C$^{-1}$ m$^{-2}$ in this temperature range.

Introduction

Line 17        I have missed a first paragraph to contextualize the topic.
Lines 17-23    This paragraph contains some information that should be included in the methods or study site section.
Line 25        A quantification of the importance of ponds as a source of CH4 to the atmosphere needs to be included.

In response to the three comments above we restructured and shortened the first paragraph of the introduction and included the first sentence of the second paragraph. We now start with the bigger picture to contextualize the topic, and quantified the pond methane emissions. The first paragraph then reads as follows:

Water bodies cover large parts of the Arctic landmasses (Muster et al., 2017), and ponds (surface area < 8 x 10$^4$ m$^2$, Ramsar et al., 2016)) are the most numerous among them (Downing et al., 2006, Polishchuk et al., 2018, Muster et al., 2019). These ponds are a relevant component within the Arctic carbon cycle (Abnizova et al., 2012), as they emit carbon dioxide and, notably, methane (Wik. et al., 2016, Holgerson et al., 2016, Beckebanze et al., 2021). In a pan-Arctic synthesis study, Kuhn et al. (2021) show that more than 30% of the total water body methane emissions come from small water bodies (area < 10$^{-1}$ km$^2$), even though they only cover 10% of the water-body area. This paper explores how pond methane emissions might change under higher temperatures.

Detailed information on the model was moved to the study section.

**Line 50**     Can not also young or recent accumulated C fuel methanogenesis?

Of course, it can, and most methanogenesis is driven by young C. We clarified this and the sentence now reads:

When the permafrost adjacent to the thawing ice wedge degrades, old carbon can leech from the thawed sediments into the pond additionally fueling methanogenesis.

**Line 54**     Add some information about how substrate quality affect CH4 emissions from ponds.

To clarify what we mean we substitute 'availability' for 'quality' and added information on the substrate:

Besides temperature, methanogenesis in water bodies depends on substrate availability. In permafrost soils, methanogens predominately use hydrogen and carbon dioxide or acetate and increasing amounts of these substrates in the soil increase the rate of methanogenesis (de Jong et al., 2018). Vascular plants are one source of substrate (Joabsson et al., 2001, Rehder et al., 2021) …

**General**     I would include a brief information about the model at the end of the introduction. Mentioning if there are other models that have been built, or why the MeEP is relevant and its novelty.

We rephrased the last paragraph to better highlight the strengths of MeEP.

To explore how pond methane emissions might change in a warmer Arctic and analyze as many of these interlinked effects on methane cycling in a single study as possible, we employ the model MeEP (Methane Emissions from Ponds). MeEP is the first model specifically developed to represent the distinct ponds of the polygonal tundra on the landscape scale (here about 5 $km^2$) and, notably, includes plant-mediated transport in addition to diffusion and ebullition.

**Material and Methods**

- MeEP seems to include vertical fluxes of water. What about lateral fluxes from the catchment? How would the model deal with those?

At the moment, MeEP does not include catchment fluxes. Since the water table is computed in a separate module, lateral fluxes could be added to the water column if they are known. Since water table variations have the smallest impact among the factors that we examine and vertical fluxes would only have a secondary impact on the water table variation we believe that the error we introduce by neglecting lateral fluxes is acceptable.

- Do the authors have some measurements of water temperature or other physic-chemical parameters to validate the assumption of well—mixed waters in summer? I ask this because I see that the model provides measurements every 1hours, while it takes half a day having a well-mixed pond.

It is important to keep in mind that the assumption that the pond is well mixed has the strongest impact on diffusion, while plant-mediated transport and ebullition are less impacted.

We use the assumption that the ponds are will mixed only for the methane module. FLake does compute the mixed-layer depths in each time step in during the open water season. FLake computes a well-mixed water column in the deeper bare part for nearly 70% of the timesteps and complete mixing at least once for over 80% of the days. Thus, methane does not have much time to accumulate in the hypolimnion even during occasional stratification. We have no dedicated measurements of the mixing state of ponds, unfortunately.

- I suggested including a sensitivity test in this section. How having different types of vegetation would affect the results? I have been in areas where ponds are moss-dominated while in other areas from the same catchment, sedges were the most dominant type of vegetation. Also, what about porosity? That links with my previous question about how general or global are the results from the model.

In terms of methane transport mosses and sedges behave very differently. In Rehder et al. (2021) we investigated impact of sedges and mosses on surface methane concentrations and found that (submerged) moss has no significant impact. So, in this study, we only focused on macrophytes. Of course, there are still differences between varying species of macrophytes which do have a big influence on methane emissions as they vary in their ability to conduct methane. However, there is very little information on the distribution of macrophyte species in our study area, which we could use for upscaling, let alone on larger scales. Thus, we do not resolve different macrophyte species and as a first step on including the impact of vegetation on water body methane emissions, include one generic type of vegetation. Please also see our comment further down with more information regarding floating moss.

Porosity mainly impacts the soil heat fluxes in MeEP and the methane only indirectly. Rather than porosity, the model is more sensitive to q10. To gauge the sensitivity of MeEP to this parameter, we run extra simulations which we discuss this in lines 221-233 and provide supplemental fig. S4.

To address the use of the model on larger scales, we added a small outlook at the end of the discussion:

Our model was set up and calibrated for one specific region featuring one specific landscape type. To quantify emissions in other regions and especially other landscape types, MeEP should be tuned with more and additional data. The magnitude of emissions depends strongly on the base productivity $P\_0$ which is the tuning parameter for the microbial communities and likely differs depending on the structure of the microbial communities. The base productivity for the vegetated pond fraction $P\_0^v$ also incorporates the impact of higher substrate availability on the microbial community. Consequently, this parameter is indirectly affected by the vegetation structure in our study region. To apply this model to other regions, special attention should be placed on availability of measurements from the overgrown parts of the ponds, especially plant-mediated transport.

Line 180:        there more studies in the area that show that these ponds are important sources of CH4?

The point we were trying to make was to highlight the comparably large area covered by ponds, less the magnitude of emissions. However, we know that surface water methane concentrations in the different pond types are similar to comparable ponds in Eastern Canada, which, in the light of upscaling of results, is in our eyes, more important than the emission strength compared to the local tundra. We added this sentence:

Surface methane concentrations in the different pond types are similar to polygonal-tundra ponds in Canada (Rehder et al., 2021).

Line 184:        What does "how compact the shape of the pond is" mean? (line 184)

Compactness measures how much a shape deviates from a perfect circle. We rephrased the sentence for simplicity.

The pond types are determined by size limits and by how circular the pond is.

Line 190        Please, justify why you assume NPP = GPP/2

NPP equals GPP minus respiration. Since respiration is notoriously difficult to model well, using a fixed ratio is a common approach in modeling. Experimental evidence suggest that biomes do tend to converge to a fixed ratio of about 0.5 (Gifford, 2003; Waring et al., 1998). We added these two citations to the manuscript.

Figure 3        The graph showns an R2 = 0.54, can the authors suggest with mechanismos could explain the rest of the variance?

One thing that is striking about the graph is that points of one color/marker pair cluster above or beyond the regression line. As each color/marker pair signify one pathway from one pond this shows that we do not resolve all the variability between the different ponds. This is also not the aim of the model. Rather we try to reproduce the average behavior, and in this sense R2 of 0.54 is reasonable. Thus, the main reason of the spread are small scale differences in the tuning parameters – not all ponds have exactly the same gas-filles porosity of base productivity of methane. We added the foll owing in the explanation about the tuning:

Much of variance can in part be explained by differences between ponds. The tuning parameters are the overall best fit, but not necessarily the best fit for each pond, so that some fluxes are over- and some are underestimated.

Results and discussion

• Line 290. What about other processes not area controlled by depth? Groundwater connectivity, for example.

How Arctic water-body areas will change in the future is a big topic and an open question which we discuss a bit further down in section 5.3. Especially in the polygonal tundra, the future landscape-scale areas are strongly impacted by thermokarst. Modeling thermokarst on the landscape scale is still work in progress, and thus we only provide a first-order estimate of the water-table dynamics based on precipitation, evaporation and above- and below-surface discharge. Permafrost is rather deep in our study region so that the ponds are not connected to groundwater from below the permafrost.

- Line 295. Is this result valid for all type of vegetation? Sedges vs mosses.

Yes. We estimate the total impact of vegetation on methane under the umbrella of overall NPP. The assumption is that more vegetation leads to more available substrate independent of the vegetation type. This is an approximation based on previous findings (Bouchard et al., 2015; Dean et al., 2020; Walter et al., 2001). While we know that ponds emit methane derived from fresh carbon, we do not have a good grasp on where this carbon comes from exactly (e.g., vegetation within versus vegetation around the pond). Thus, before attempting a more intricate modeling approach we believe that we first need to improve our understanding of the carbon pathways in ponds.

- Kuhn et al. 2018 – open water ponds, higher CH4 fluxes.

From what we gather from Kuhn et al. (2018), their focus was on ebullition and diffusion. The difference between open and overgrown water is mostly due to plant-mediated transport in our study. In line with Kuhn et al. (2018) we find lower ebullition and diffusion in the overgrown parts of the ponds compared to the open water. This difference is, however, overcompensated by strong plant-mediated transport. Additionally, the open water ponds in Kuhn et al. (2018) are those with active erosion along the shores, which leads to additional input of organic material into the pond. As mentioned in the paper and above, these erosion processes are not part of MeEP, but could exert positive feedback on methane emissions. However, in our study area, there is very little erosion at present.

- Units: mmol m-2 d-1

We are unsure which line this comment refers to but consistently used gram instead of mol in the results and discussion section as this is the more widely used unit.

- Line 305 Check Kuhn et al. 2008. Maybe authors need to be more specific and mentioning that these results are ponds from that specific area.

Since we did not find a relevant paper from Kuhn et al. (2008), we assume this also refers to the above paper Kuhn et al. (2018). Please also refer to our comment above. Notably, the moss in Kuhn et al. (2018) is very different from the moss in our study region and we added a few sentences to address this issue in the discussion:

One caveat when adapting MeEP for the larger scale is that in our study area ponds do not feature floating mosses like sphagnum which can be found at other sites and

reduce methane emissions (Kuhn et al., 2018). While submerged mosses do not impact surface methane concentrations in our study site (Rehder et al., 2021), the same might not be true for floating vegetation.

Bouchard, F., Laurion, I., Pr, x, skienis, V., Fortier, D., Xu, X. and Whiticar, M.J., 2015. Modern to millennium-old greenhouse gases emitted from ponds and lakes of the Eastern Canadian Arctic (Bylot Island, Nunavut). Biogeosciences, 12(23): 7279-7298.

Dean, J.F., Meisel, O.H., Martyn Rosco, M., Marchesini, L.B., Garnett, M.H., Lenderink, H., van Logtestijn, R., Borges, A.V., Bouillon, S., Lambert, T., Röckmann, T., Maximov, T., Petrov, R., Karsanaev, S., Aerts, R., van Huissteden, J., Vonk, J.E. and Dolman, A.J., 2020. East Siberian Arctic inland waters emit mostly contemporary carbon. Nature Communications, 11(1): 1627.

Gifford, R.M., 2003. Plant respiration in productivity models: conceptualisation, representation and issues for global terrestrial carbon-cycle research. Functional Plant Biology, 30(2): 171-186.

Rehder, Z., Zaplavnova, A. and Kutzbach, L., 2021. Identifying Drivers Behind Spatial Variability of Methane Concentrations in East Siberian Ponds. Frontiers in Earth Science, 9(183).

Walter, B.P., Heimann, M. and Matthews, E., 2001. Modeling modern methane emissions from natural wetlands 1. Model description and results. Journal of Geophysical Research-Atmospheres, 106(D24): 34189-34206.

Waring, R., Landsberg, J. and Williams, M., 1998. Net primary production of forests: A constant fraction of gross primary production? Tree Physiology, 18: 129-134.

---

## Author Comment (AC2)

Dear reviewer,

Thank you very much for your positive and constructive feedback and comments on our manuscript. We tried to incorporate your suggestions into the paper. Please find below your comments in orange and our point-by-point response in black. Updated excerpts of the manuscript are included in blue.

The manuscript by Rehder et al. focuses on the simulation of methane emissions from tundra ponds. The authors tackle an important topic given the unprecedented warming in this part of the world. Consequently, the abundance of such ponds could become even larger in the future. In order to simulate methane emissions the authors classified three types of ponds and applied a process-based model. The tuning of the model was achieved by using previously collected data in the Lena Delta - I thoroughly studied region in the Arctic. Furthermore the model is used to estimate methane emissions with ongoing warming.

The paper is well written and concise and the results are sound. Particularly the classification in different poond types as well as the contribution of different pathways and their subsequent attribution is intriguing. The major concern is the calibration of the model with this single site. I am aware that data, and particularly flux data from ponds are not widely available, however a detailed discussion on how different soil types or vegetation structure would affect the fluxes is necessary. The authors themselfes mention at the beginning of the manuscript (p4, l86) that this is a first order approximation for sandy and organic-rich sediments. Surely this is not the case for many other regions. There are some hints towards the upscaling of the results in the conclusion, however a distinct section on how the model can be used and particularly what is needed to achieve upscaling - more precisely what this would mean for the fluces at regional scale - would be very beneficial.

Thank you for your comment. To briefly summarize: (1) You wonder how dependent on soil type/vegetation structure the calibration is. Linking this to upscaling (2) you would like to see a section discussing the applicability of the model.

Regarding (1), the way MeEP was set up, the soil type will have a stronger influence on the thermal structure of the pond and only indirectly influence methane emissions. However, the soil type might influence the microbial community, and in this way the base productivity, one of our tuning parameters. When applying MeEP to larger regions, in a first step an average $P_0^{v/b}$ determined with measurements from several regions will already give new insights on the impact of vegetation on methane emissions from small waterbodies, especially when paired with information about the overgrown area of ponds. So far, plant-mediated fluxes have not been considered when upscaling waterbody emissions, so even a first estimate that does not resolve all regional differences would be a step forward. We added a paragraph on using MeEP for other or larger regions at the end of the discussion:

Our model was set up and calibrated for one specific region featuring one specific landscape type. To quantify emissions in other regions and especially other landscape

types, MeEP should be tuned with more and additional data. The magnitude of emissions depends strongly on the base productivity $P_0$ which is the tuning parameter for the microbial communities and likely differs depending on the structure of the microbial communities. The base productivity for the vegetated pond fraction $P_0^v$ also incorporates the impact of higher substrate availability on the microbial community. Consequently, this parameter is indirectly affected by the vegetation structure in our study region. To apply this model to other regions, special attention should be placed on availability of measurements from the overgrown parts of the ponds, especially plant-mediated transport. One caveat when adapting MeEP for the larger scale is that in our study area ponds do not feature floating mosses like sphagnum which can be found at other sites and reduce methane emissions (Kuhn et al., 2018). While submerged mosses do not impact surface methane concentrations in our study site (Rehder et al., 2021), the same might not be true for floating vegetation.

Two additional specific comments:

I was missing a clear research question and hypothesis

We changed the last paragraph of the introduction, with the research question in the first sentence of the paragraph –

We aim to **explore how pond methane emissions might change in a warmer Arctic** and analyze as many of these interlinked effects on methane cycling in a single study as possible by employing the model MeEP (Methane Emissions from Ponds).

– and the hypothesis in the last sentence of the same paragraph:

While diffusion and ebullition are usually accounted for, the impact of plant-mediated transport on landscape-scale fluxes from ponds is usually not considered but **we expect it to be as important as the other two fluxes**.

Figure 9: the combination of the area in panel b does not necessarily relate to panel a, Also in the caption you write about river terraces, yet in the figure nothing about these is mentioned

Yes, panel (a) is per area of landcover type, panel (b) per area of polygonal tundra. To clarify, we slightly adapted the caption (and substituted 'river terrace' with 'polygonal tundra', a word we use more often in the manuscript.) The caption now reads:

Impact of pond emissions on landscape methane emissions. (a) **For the hist_all simulation, we compare fluxes from different landscape elements.** The estimate for the overall tundra emissions **(orange bar)** were acquired with eddy-covariance measurements over the growing season of 2003 (Wille et al., 2008) **and are shown for comparison.** Note, that the influence of ponds on these measurements is low. The methane emissions per square meter of open and overgrown water are broken down per pond type. (b) Methane emissions per square kilometer of **polygonal tundra** of each pond type are displayed as stacked bars. We compare these emissions per pond type to the area this pond type covers in the **polygonal tundra** of Samoylov Island (sand-colored bar). This comparison relies on the assumption that the

emissions measured by (Wille et al., 2008) are representative for **polygonal-tundra** emissions.

I hope these comments are useful and I enjoyed reading the manuscript.

Thank you very much. The comments are very helpful.

Bouchard, F., Laurion, I., Pr, x, skienis, V., Fortier, D., Xu, X. and Whiticar, M.J., 2015. Modern to millennium-old greenhouse gases emitted from ponds and lakes of the Eastern Canadian Arctic (Bylot Island, Nunavut). Biogeosciences, 12(23): 7279-7298.

Dean, J.F., Meisel, O.H., Martyn Rosco, M., Marchesini, L.B., Garnett, M.H., Lenderink, H., van Logtestijn, R., Borges, A.V., Bouillon, S., Lambert, T., Röckmann, T., Maximov, T., Petrov, R., Karsanaev, S., Aerts, R., van Huissteden, J., Vonk, J.E. and Dolman, A.J., 2020. East Siberian Arctic inland waters emit mostly contemporary carbon. Nature Communications, 11(1): 1627.

Gifford, R.M., 2003. Plant respiration in productivity models: conceptualisation, representation and issues for global terrestrial carbon-cycle research. Functional Plant Biology, 30(2): 171-186.

Rehder, Z., Zaplavnova, A. and Kutzbach, L., 2021. Identifying Drivers Behind Spatial Variability of Methane Concentrations in East Siberian Ponds. Frontiers in Earth Science, 9(183).

Walter, B.P., Heimann, M. and Matthews, E., 2001. Modeling modern methane emissions from natural wetlands 1. Model description and results. Journal of Geophysical Research-Atmospheres, 106(D24): 34189-34206.

Waring, R., Landsberg, J. and Williams, M., 1998. Net primary production of forests: A constant fraction of gross primary production? Tree Physiology, 18: 129-134.